# Anti-Neutrophil Cytoplasmic Antibody-Associated Vasculitis in Kidney Transplantation

**DOI:** 10.3390/medicina57121325

**Published:** 2021-12-03

**Authors:** Valentina Binda, Evaldo Favi, Marta Calatroni, Gabriella Moroni

**Affiliations:** 1Nephrology, Dialysis and Transplantation, Fondazione IRCCS Ca’ Granda Ospedale Maggiore Policlinico, 20122 Milan, Italy; valentina.binda@policlinico.mi.it; 2Kidney Transplantation, Fondazione IRCCS Ca’ Granda Ospedale Maggiore Policlinico, 20122 Milan, Italy; 3Department of Clinical Sciences and Community Health, University of Milan, 20122 Milan, Italy; 4Department of Biomedical Sciences, Humanitas University, 20089 Rozzano, Italy; marta.calatroni@hunimed.eu (M.C.); gabriella.moroni@hunimed.eu (G.M.); 5IRCCS Humanitas Research Hospital, 20089 Rozzano, Italy

**Keywords:** chronic kidney disease, kidney transplant, ANCA-associated vasculitis, pauci-immune glomerulonephritis, patient survival, graft survival, outcome, review

## Abstract

Due to complex comorbidity, high infectious complication rates, an elevated risk of relapsing for primary renal disease, as well as inferior recipient and allograft survivals, individuals with anti-neutrophil cytoplasmic antibody (ANCA)-associated vasculitis (AAVs) are often considered as poor transplant candidates. Although several aspects of recurrent and de novo AAVs remain unclear, recent evidence suggests that kidney transplantation (KT) represents the best option, which is also the case for this particular subgroup of patients. Special counselling and individualized approaches are strongly recommended at the time of enlistment and during the entire post-transplant follow-up. Current strategies include avoiding transplantation within one year of complete clinical remission and thoroughly assessing the recipient for early signs of renal or systemic vasculitis. The main clinical manifestations of allograft AAV are impaired kidney function, proteinuria, and hematuria with ANCA positivity in most cases. Mixed results have been obtained using high-dose steroids, mycophenolate mofetil, or cyclophosphamide. The aim of the present review was to summarize the available literature on AAVs in KT, particularly focusing on de novo pauci-immune glomerulonephritis.

## 1. Survival of Patients with Anti-Neutrophil Cytoplasmic Antibody-Associated Vasculitis

Anti-neutrophil cytoplasmic antibody (ANCA)-associated vasculitis (AAVs) represents a heterogeneous group of small-vessel vasculitis including systemic forms such as granulomatosis with polyangiitis (GPA), microscopic polyangiitis (MPA), and eosinophilic granulomatosis with polyangiitis (EGPA or Churg–Strauss syndrome) as well as forms limited to the kidney, also known as renal-limited vasculitis (RLVs) [1,2]. Therefore, the spectrum of AAVs may range from isolated and indolent single-organ involvement to life-threatening fulminant disease.

Until recently, the prognosis of untreated patients with AAVs has been extremely poor, with mortality rates as high as 80% within the first year of diagnosis and most of the deaths occurring due to renal or respiratory failure [3]. Over the years, tremendous advances in the understanding of the pathophysiology of AAVs and the development of more effective therapeutic strategies have led to a significant amelioration of results [4]. Nonetheless, long-term outcomes are still suboptimal. GPA and MPA have been associated with 5-year survival rates of 74–79% and 46–80%, respectively [5,6,7,8,9]. A large meta-analysis of observational studies including 3338 affected subjects concluded that there was a 2.7-fold increase in mortality among patients with AAVs compared with the general population, regardless of sex [10]. The risk of death is highest during the first year of follow-up, but excess mortality remains persistently elevated [11]. The leading causes of death in the short term are infections and active vasculitis, whereas long-term mortality is predominantly determined by cardiovascular disease (CVD), malignancy, or infections [5]. Remarkably, long-term survival may be further compromised by drug-induced side effects and associated comorbidities [8]. Recognized risk factors for adverse outcomes are severity of disease at onset, age at diagnosis, number of relapses, and duration of steroid administration [12].

The aim of the present review was to summarize current knowledge on AAVs in the KT setting, particularly focusing on the differences between recurrent and de novo forms of the disease. Most relevant follow-up and management strategies are also highlighted.

## 2. End-Stage Renal Disease in ANCA-Associated Vasculitis

Renal vasculitis is a frequent manifestation of GPA and MPA, occurring in 50% of patients at onset and in up to 70–85% during the course of the disease [13,14]. On the contrary, severe involvement of the kidney is seldom observed in EGPA. It has been estimated that a quarter of individuals with AAVs eventually develop end-stage renal disease (ESRD) within five years of diagnosis [13,15,16,17]. In particular, early renal replacement therapy (RRT) is often required in GPA and MPA patients presenting with rapid and progressive kidney failure. Among these subjects, one third actually fails to respond to treatment, thus requiring life-long dialysis or kidney transplantation (KT) [18]. Progressive deterioration of renal function can also be observed in patients with mild-to-moderate involvement of the kidney as a result of glomerular, tubular, or vascular scarring. The occurrence of further episodes of renal vasculitis in association with systemic arterial hypertension, atherosclerosis, or diabetes may act as an important contributing factor.

There is a lack of information regarding AAVs patients with chronic kidney disease (CKD), especially those on RRT. Analyzing data from 302 subjects with AAVs (MPA, 136 and GPA, 167), enrolled in six European Vasculitis Society (EUVAS) trials and followed-up for 7.1 years, Robson and colleagues estimated a 14% incidence of ESRD [19]. In a Swedish observational cohort study with a 5-year follow-up and including 201 new cases of AAVs (PR3-ANCA, 98 and MPO-ANCA, 85) diagnosed between 1997 and 2009, 20% of patients developed ESRD. The proportion of individuals requiring dialysis was significantly higher in the group with MPO-ANCA (38%) than PR3-ANCA (5%) [20]. Lionaki et al. retrospectively assessed the outcomes of 532 AAVs patients. Over a median observation time of 40 months, ESRD was recorded in 136 participants (26%). Reasons for kidney failure were new-onset AAVs (51%), progressive CKD (43%), and renal relapse of pre-existing AAVs (6%) [21]. According to the Renal Epidemiology and Information Network (REIN) registry, between 2002 and 2011, 425 patients with AAVs (MPA, 166 and GPA, 259) started chronic RRT, accounting for 0.7% of all the incident dialysis population. As previously reported by other studies [21,22], hemodialysis was the preferred modality (94%), predominantly using a central line catheter (66%). The median age at initiation of RRT was 70 years (range 7–93) [23].

Overall, available data do not show substantial differences in crude survival rates between chronic dialysis patients with AAVs and those with other primary renal diseases [23,24,25]. Conversely, the outcomes of KT, in this particular subgroup of candidates, remain controversial.

## 3. Kidney Transplantation in Patients with ANCA-Associated Vasculitis

It has been estimated that only 14% to 22% of the subjects with AAVs and ESRD eventually receive a KT, regardless of sex or age at the time of enlistment. Such low numbers are sadly similar to those reported in Italy or France for patients aged 70 years or more [23,24,26]. According to the Unites States Renal Data Service (USRDS), among patients with primitive or secondary glomerulonephritis on chronic RRT, those with AAVs exhibit the lowest transplantation rate (3.1 events for 100 patient-year) and the highest mortality rate (13.2 events for 100 patient-year) [27]. Using the aforementioned registry, Wallace and colleagues showed that only 15.9% (*n* = 946) of the 5929 patients diagnosed with GPA-related ESRD between 1995 and 2014 were actually transplanted in the following years. Overall, 438 deaths were recorded during the follow-up: 199 in the KT group and 239 in the dialysis group. The mortality rates for transplanted and non-transplanted patients were 29.3/1000 patient-year and 65.5/1000 patient-year, respectively. On multivariable analysis, KT was associated with a 70% reduction in the risk of death, particularly from CVD [28]. Accordingly, in a very recent work, it was observed that AAVs patients undergoing KT were less likely to develop myocardial infarction and ischemic stroke than their dialysis counterpart [29]. Positive results were also observed outside the United States. In a study analyzing data from 12 European renal registries and assessing long-term outcomes of 558 KT recipients with AAVs, 10-year patient and graft survival rates as high as 74.8% and 63.7% were reported. Remarkably, using multiple Cox regression models and following adjustments for time periods and countries, no significant differences in mortality could be detected between AAVs recipients and matched control groups with primary glomerulonephritis or other renal diseases [26]. Ten-year patient survival rates ranging from 65% to 87% as well as 10-year graft survival rates between 64% and 84% were described by other authors, in smaller series [30,31,32,33]. In particular, equivalent long-term patient- and transplant-related outcomes were reported by Moroni et al. [33] and by Marco et al. [32] comparing KT recipients with or without AAVs. Nevertheless, in one of these studies, infectious complications were significantly higher in the AAVs group (74% vs. 34%; *p* = 0.01) [33].

Even though there is now evidence that AAVs do not represent an absolute contraindication to KT [32,34,35,36], extra caution and specific counselling are advised, especially for potential recipients with MPA. A large retrospective study assessing the outcomes of MPA (*n* = 46), GPA (*n* = 47), or non-AAV (*n* = 8193) patients who had been transplanted in Australia and New Zealand between 1996 and 2010 showed that MPA was associated with lower 10-year graft (MPA, 50% vs. GPA, 62% vs. non-AAV, 70%) and recipient (MPA, 68% vs. GPA, 85% vs. non-AAV, 92%) survivals than controls. On multivariable regression analysis, the risk of premature transplant failure was almost equivalent in GPA and non-AAV recipient0s, whilst it was 1.87 times higher in the MPA group [24]. Albeit limited to a single report, a discrepancy in the incidence of post-transplant fatal cardiac events and malignancy was also noticed between MPA and GPA patients [24].

## 4. Timing of Kidney Transplantation in Patients with ANCA-Associated Vasculitis

The optimal timing of KT in patients with AAVs is still debated. However, there is a consensus that transplantation within 12 months of disease remission is associated with a strikingly elevated risk of death. As demonstrated by Little et al. in a population of 107 subjects, being transplanted before one year has passed since AAV remission represents the strongest predictor of recipient death (hazard ratio, 2.3; *p* < 0.05) [30]. Accordingly, the Kidney Disease Improving Global Outcomes (KDIGO) guidelines [37] and the Canadian Society of Transplantation [38] both recommend to wait one year after clinical remission before enlistment.

Although circulating ANCA does not represent a reliable indicator of active disease, persistently elevated ANCA levels, especially PR3-ANCA, are a well-recognized risk factor for AAVs relapse in the general population [36]. In the KT setting, the relationship between ANCA positivity and AAVs recurrence is less clear. Some single-center studies reported that abnormal ANCA concentrations were not associated with post-transplant relapsing AAVs [30,39]. On the contrary, analyzing a larger population, Marco et al. found that ANCA-positive patients at the time of transplant were more likely to experience AAVs relapse than those with normal ANCA levels (17% vs. 5%, respectively) [32]. Remarkably, the risk of post-transplant relapse seems to be highest in recipients with PR3 ANCA positivity [40].

Considering the scarcity of data currently available and acknowledging the limitations of ANCA testing as a diagnostic biomarker of relapsing AAVs [41], it sounds reasonable to follow the latest KDIGO guidelines [42] and avoid delaying a KT in patients with sustained clinical remission and ANCA positivity. Waiting for stronger evidence, a cautious and individualized approach is suggested, including aggressive clinical follow-up and routine ANCA levels monitoring before and after transplant [36].

## 5. ANCA-Associated Vasculitis Relapse after Kidney Transplantation

AAVs relapse rate after KT is generally low, ranging between 0.003 and 0.076 per patient per year [24,33]. The average time from transplant to recurrence is 30.9 months, but it can be extremely variable (from 15 days to 16 years) [43].

Patients with GPA are more likely to experience post-transplant AAV relapse than those with MPA [32,44,45], and recurrence rates are higher in subjects with positive ANCA at transplant [46]. The duration of primary vasculitis, the pattern of circulating ANCA, previous treatments with cyclosporine, the length of RRT, or the donor type do not seem to affect the risk of relapse after transplant [31,33,34,47,48]. In particular, no significant differences in recurrence rates were observed between patients with or without ANCA positivity at the time of transplant or between recipients with different circulating ANCA subtypes [43,49]. Episodes of relapsing AAVs with undetectable ANCA were also reported [49].

It has been estimated that about 60% of recurrent AAVs involve the transplanted kidney, alone or in combination with other organs. Isolated extra-renal involvement occurs in 40% of cases [43]. Microscopic hematuria and proteinuria represent heralding signs of graft relapse. These manifestations are generally associated with or shortly followed by impaired renal function [33]. Histologic features mostly resemble those observed in the native kidney, characterizing a focal or diffuse pauci-immune extra-capillary necrotizing glomerulonephritis in acute phases. Assessing ten transplant biopsies from patients with relapsing AAVs and using the classification proposed by Berden et al. [50], it was found that five patterns were focal, four mixed, and one crescentic [49].

The most frequently used therapeutic protocol for post-transplant recurrent AAVs includes steroid pulses, cyclophosphamide, and plasmapheresis with a concomitant reduction in anti-rejection prophylaxis [51]. Successful treatment with rituximab has been reported [50].

The long-term impact of AAVs recurrence after KT remains unclear. Evaluating data from the Australia and New Zealand Dialysis and Transplant Registry (ANZDATA), Briganti et al. demonstrated that post-transplant relapsing AAVs were associated with a 10-year graft loss rate of 7.7% [52]. However, other studies suggest less-optimistic outcomes, showing early transplant loss in more than 30% of the patients [33,50,51]. Independent predictors of AAVs recurrence-related transplant failure are graft involvement and male sex [50].

Most relevant literature on relapsing AAVs in KT is summarized in Table 1.

## 6. De Novo Pauci-Immune Glomerulonephritis after Kidney Transplantation

New-onset AAV after KT is an extremely rare condition. To date, little is known about incidence, risk factors, and long-term outcomes of transplant recipients developing de novo pauci-immune glomerulonephritis (PIGN) [53,54].

The first report on a case of de novo kidney allograft ANCA-PIGN was written by Asif in 2000 [55]. A 52-year-old female deceased donor KT recipient with no history of autoimmune disease was hospitalized due to malaise, arthralgias, mild headache, diplopia, and sores on the left eye. At the time of admission, the patient was on a maintenance immunosuppressive scheme consisting of cyclosporine and steroids. Laboratory tests showed increased serum creatinine concentration (SCr), moderate proteinuria, and 50–100 red blood cells (RBC)/high-power field (HPF) in the urinary sediment. Histology with immunofluorescence staining demonstrated a necrotizing vasculitis involving the glomerular capillaries with crescents formation and no immune complexes. *p*-ANCA and MPO-ANCA were both elevated. The patient managed stopping cyclosporine and administering intravenous (IV) methylprednisolone (1 gr for three days) and oral cyclophosphamide (125 mg a day). After six months of treatment, graft function remained stable (SCr, 4 mg/dL) without signs of clinically active disease. Another case of post-transplant de novo ANCA-PIGN was described by Tabata et al. in 2009 [56]. A 34-year-old woman with ESRD secondary to IgA nephropathy, who had undergone living donor KT at the age of 21, was admitted after an uneventful course due to a slight increase in SCr, microscopic hematuria, and mild proteinuria. A graft biopsy revealed that 8/22 glomeruli were globally sclerotic, whereas 9/22 exhibited tuft necrosis with cellular crescents. Immunofluorescence staining was negative for mesangial deposition of C3 or IgA. Circulating MPO-ANCA levels were minimally elevated. The patient was treated with IV methylprednisolone (500 mg for three days) and oral prednisone (0.6 mg/kg/day), achieving partial remission of the disease. She eventually lost her graft and returned to dialysis five years after diagnosis. Further information about post-transplant de novo ANCA-PIGN were obtained by Haruyama et al. [57]. A 61-year-old female living donor KT recipient with ESRD secondary to chronic glomerulonephritis and 31 years of follow-up was admitted to investigate an unexpected raise in SCr (from 0.6 to 1.39 mg/dL) associated with abnormal urinary protein/creatinine ratio (1.39 g/g) and microscopic hematuria (>100 RBC/HPF). Histological evaluation demonstrated crescentic PIGN with moderate to severe tubulointerstitial inflammation. Circulating MPO-ANCA levels were 45.5 U/mL. Following treatment with IV methylprednisolone (500 mg for three days) and oral prednisone, complete clinical remission and restored graft function were obtained. A case of de novo PR3-ANCA-PIGN was reported in a 66-year-old asymptomatic woman who had undergone deceased donor KT due to autosomal dominant polycystic kidney disease [58]. After 20 months of follow-up, routine laboratory tests showed elevated SCr (from 1.8 to 2.6 mg/dL), abnormal proteinuria (urinary protein/creatinine ratio, 0.44 mg/g), and microscopic hematuria (20 RBC/HPF). A graft biopsy demonstrated PIGN with extra-capillary proliferation and interstitial inflammation. Highly elevated circulating PR3-ANCA were also detected. At first, the subject was given IV methylprednisolone (250 mg for three days), rituximab (1 gr total dose), and oral prednisone (50 mg/day). The clinical response was poor with worsening renal function and histological evidence of persistent disease. Therefore, a rescue therapy with seven sessions of plasmapheresis was attempted. Following treatment, a significant decrease in PR3-ANCA levels and transient amelioration of renal function were observed. However, the course was complicated by multiple infections, eventually leading to transplant loss in a few months. A study describing ten KT recipients with de novo ANCA-PIGN was recently published by Buglioni et al. [59]. The mean time from transplant to disease diagnosis was 32 months (range, 4–96 months). The main clinical manifestations were elevated SCr (10/10 patients; mean, 3.59 mg/dL; range, 1.3–8 mg/dL), abnormal proteinuria (positive, *n* = 5; negative, *n* = 2; undetermined, *n* = 3), and microscopic hematuria (positive, *n* = 5; negative, *n* = 2; undetermined, *n* = 3). There were no patients exhibiting signs of systemic involvement. All graft biopsies showed features of focal necrotizing or crescentic glomerulonephritis (mean glomerular involvement, 16%; range, 2–36%) with a pauci-immune pattern at immunofluorescence and electron microscopy. Seven recipients were assessed for circulating ANCA; PR3-ANCA were detected in one. Information regarding treatment was available for six patients. The therapeutic strategies were as follows: prednisone administration (*n* = 3), increased dosage of mycophenolate mofetil (*n* = 1), a switch from azathioprine to mycophenolate mofetil (*n* = 1), and no modifications (*n* = 1). Within one year of diagnosis, six recipients still had a functioning graft, thus suggesting that post-transplant de novo ANCA-PIGN may be less aggressive than AAVs involving the native kidney. In this regard, it has been speculated that the rarity and the relatively mild course of the disease observed in KT recipients reflect the extremely low incidence recorded in the general population (13–20 million-year) as well as the protective effect of post-transplant induction and maintenance immunosuppression [41].

The reported literature on de novo ANCA-PIGN after KT is summarized in Table 2.

## 7. Conclusions

In the last two decades, there has been a remarkable improvement in the understanding and management of AAVs. Despite these encouraging results, overall outcomes remain disappointing. As a matter of fact, patients with systemic manifestations of the disease exhibit higher mortality rates than the general population, and about one third of those with renal involvement eventually require chronic RRT [10].

Due to complex comorbidity, higher infectious complication rates, the risk of recurrent primary renal disease, as well as inferior long-term recipient and graft survival, individuals with AAVs have often been considered as poor transplant candidates [30,60,61]. There is now evidence that KT is also the treatment of choice in this particular subgroup of patients as it is associated with superior quality of life and longer life expectancy [36,40].

Considering the heterogeneity of the disease and the limited amount of data available, special counselling and individualized approaches are warranted at the time of enrolment in the transplant waiting list. In particular, international guidelines and transplant societies recommend to postpone the transplant procedure until one year has passed from complete clinical remission and to use extra caution in patients previously exposed to intense immunosuppressive protocols [30,42,60,61].

Relapsing AAV occurs in up to 11% of KT recipients, and it is frequently associated with severe complications and premature transplant loss [32]. Although extremely rare, post-transplant de novo ANCA-PIGN is challenging [53]. The time of onset of relapsing and de novo AAVs are greatly variable and predisposing factors are still undetermined. Diagnosis is frequently incidental, and the course of the disease can be particularly insidious. The main clinical manifestations include worsening graft function, abnormal proteinuria, and microscopic hematuria. In our experience, monitoring urine analysis (i.e., protein/creatine ratio and RBC count) every month during the first year after transplant and every three months thereafter as well as checking for circulating ANCA every six to twelve months represents an effective and affordable follow-up strategy [33]. A low threshold for graft biopsy is also advised as prompt initiation of treatment may improve the response rate and outcomes, especially in de novo ANCA-PIGN [59].

To date, there is not a standardized treatment for relapsing AAVs or de novo PIGN [33,50,59,62,63]. In general, post-transplant recurrent AAVs seem to be more aggressive than de novo ANCA PIGN, but the latter has been associated with lower response rates. Mycophenolate mofetil-containing maintenance immunosuppressive regimens are probably the best option as they can effectively act on both rejection and relapse. In case of recurrence, first-line treatment should include steroids and cyclophosphamide, reserving apheresis and rituximab for non-responders [64].

## Figures and Tables

**Table 1 medicina-57-01325-t001:** Most relevant literature on relapsing anti-neutrophil cytoplasmic antibody-associated vasculitis after kidney transplantation.

Authors Year Reference	Study Design	Patients (*n*)	Relapse (*n*/%)	Relapse (Patient-Y)	KT to Relapse (Months)	Involvement (*n*)	ANCA (*n*)	Treatment (*n*)	Outcomes (*n*)
Nachman et al., 1996 [15]	retrospective multi-center	127	22 (17.3%)	0.070	30.9 (4–89)	renal (12) extra-renal (10)	c-ANCA (9) *p*-ANCA (5)	CYC (12) AZA (3) MP (1)	death (1) graft loss (3) remission (11)
Tang et al., 2013 [24]	retrospective multi-center	93	2 (2.1%)	0.003	79.8 and 57.4	renal (2)	N/A	N/A	graft loss (1) remission (1)
Little et al., 2009 [30]	retrospective multi-center	107	5 (4.7%)	0.010	N/A	N/A	N/A	N/A	death (1) graft loss (3)
Geetha et al., 2011 -[31]	retrospective multi-center	85	8 (9.4%)	0.020	3–55	renal (4) extra-renal (4)	N/A	CYC (1) AZA→MMF (1)	death (1) graft loss (1)
Marco et al., 2013 [32]	retrospective multi-center	49	3 (8.1%)	0.010	62 (1–109)	renal (2) extra-renal (1)	MPO (2) PR3 (1)	MP (3) CYC (1)	death (1) graft loss (2)
Moroni et al., 2007 [33]	retrospective single-center	19	7 (37%)	0.076	45 (0.5–192)	renal (7)	N/A	MP + CYC (7)	death (1) graft loss (2) remission (3)
Shen et al., 2011 [34]	retrospective multi-center	919	12 (1.3%)	0.003	45 (0.5–116)	renal (12)	N/A	N/A	graft loss (7)
Göçeroğlu et al., 2016 [49]	retrospective multi-center	113	13 (12%)	0.030	28.2 (0.3–59.2)	renal (11) extra-renal (2)	MPO (5) PR3 (6) N/A (2)	CYC (8) RTX (2) AZA (1) P (1)	graft loss (4) remission (9)

Abbreviations: KT, kidney transplant; ANCA, anti-neutrophil cytoplasmic antibody; CYC, cyclophosphamide; AZA, azathioprine; MP, methylprednisolone; N/A, not available; MMF, mycophenolate mofetil; RTX, rituximab; P, prednisone.

**Table 2 medicina-57-01325-t002:** Reported cases of de novo pauci-immune glomerulonephritis after kidney transplantation.

Authors Year Reference	Sex (*n*)	Age at KT (Years)	Donor	KT to PIGN (Years/Months)	PRD (*n*)	ANCA (*n*)	Treatment (*n*)	Outcomes (#)
Asif et al., 2000 [55]	F	38	DD	14 years	Unknow	MPO	MP pulses CYC	Impaired graft function (6 months)
Tabata et al., 2009 [56]	F	19	LD	13 years	IgA-N	MPO	MP pulses *p*	ESRD (5 years)
Haruyama et al., 2015 [57]	F	30	LD	31 years	Chronic GN	MPO	MP pulses *p* (30 mg/day)	Stable renal function (3 months)
Sagmeister et al., 2018 [58]	F	65	DD	20 months	ADPKD	PR3	MP pulses RTX (1 g) PEX	ESRD (4 months)
Buglioni et al., 2020 [59]	F (5) M (5)	51 (26–78)	N/A	32 months (4–96)	SLE (3) ADPKD (2) DM (3) FSGS (1) IN (1)	No ANCA (6) c-ANCA (1) N/A (3)	P (3) MMF (1) AZA→MMF (1) no change (1)	ESRD (2) death with function (3) stable graft function (3)

Abbreviations: PIGN, pauci-immune glomerulonephritis; KT, kidney transplant; PRD, primary renal disease; ANCA, anti-neutrophil cytoplasmic antibody; F, female; DD, deceased donor; MP, methylprednisolone; CYC, cyclophosphamide; LD, living donor; IgA-N, IgA nephropathy; P, prednisone; ESRD, end-stage renal disease; GN, glomerulonephritis; ADPKD, autosomal dominant polycystic kidney disease; RTX, rituximab; PEX, plasmapheresis; M, male; N/A, not available; FSGS, focal segmental glomerulosclerosis; SLE, systemic lupus erythematosus; DM, diabetes mellitus; IN, interstitial nephritis; AZA, azathioprine; MMF, mycophenolate mofetil.

## Data Availability

No new data generated.

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
