# Peer review of "Anti-Neutrophil Cytoplasmic Antibody-Associated Vasculitis in Kidney Transplantation"

_medicina, 2021, doi:10.3390/medicina57121325_

Round 1

Reviewer 1 Report

Manuscript ID: medicina-1458114

“Anti-neutrophil cytoplasmic antibody-associated vasculitis in kidney transplantation”

This review aims to summarize available literature on AAVs in KT, particularly focusing on de novo pauci-immune glomerulonephritis. The review was carefully done. I think that this manuscript is interesting. However, there are some flaws, which should be resolved as following.

  1. Systematic reviews and meta-analyses are considered to be the highest quality evidence on a research topic because their study design reduces bias and produces more reliable findings. Please add evidence from recent systematic review and meta-analysis.
  2. More than a half number of references were published more than 5-10 years. Please update citation from a recent standard peer review journal.
  3. It is important that within the manuscript, the authors clarify the importance of this work, how it differs from and advances previously published work and how this article can benefit the field and patients in the future etc. Please also add more information from recently published research and offer a more speculative and forward-looking perspective.
  4. Finally, since I am not a native English user, I did not check for grammatical errors thoroughly. This should be done by an appropriate language reviewer.

Author Response

We would like to thank Reviewer 1 for his/her comments and suggestions.

Q1) Systematic reviews and meta-analyses are considered to be the highest quality evidence on a research topic because their study design reduces bias and produces more reliable findings. Please add evidence from recent systematic review and meta-analysis.

A1) We certainly agree on the fact that systematic reviews and meta-analysis offer more reliable information than a narrative review. However, available literature on the particular topic of our manuscript is quite limited. Accordingly, there are few systematic reviews or meta-analysis, especially published in recent years. In order to address the comment, we have cited two more papers (lines 130-133, ref. 62 and lines 155-157, ref. 65).

Q2) More than a half number of references were published more than 5-10 years. Please update citation from a recent standard peer review journal.

A2) As previously mentioned, available literature on the topic is scarce and most papers were published more than ten years ago. In an attempt to address the comment, we have further checked PubMed and Scopus for any recent manuscript. As a result, four extra articles have been included in the revised version of the review:  I] lines 5-6, ref. 29; II] lines 47-49, ref. 40; III] lines 130-133, ref. 62; and IV] lines 155-157, ref. 65.

Q3) It is important that within the manuscript, the authors clarify the importance of this work, how it differs from and advances previously published work and how this article can benefit the field and patients in the future etc. Please also add more information from recently published research and offer a more speculative and forward-looking perspective.

A3) As a narrative review, we are aware that our contribution cannot add any relevant new insight to the topic. However, considering the rarity of the disease and the mixed results reported by most of the studies, we think that a more comprehensive discussion, particularly regarding possible differences between relapsing and de novo AAVs, could offer an easy to use reference for counselling and management of this rather complex group of patients. Such statement has been included in the introduction section of the revised version of the manuscript.

Reviewer 2 Report

This is an interesting summary of ANCA vasculitis in Kidney Transplant recipients.  There is no new information here, so a more thorough summary of the literature referenced may be more helpful to the reader.

What evidence is there for waiting one year after the resolution of AAV symptoms?  Do any of your references demonstrate the advantage of waiting one year?

Pages 2/3 of 12: you state the median age of RRT in AAV renal failure was 70, and then you state this group had a low transplant rate.  You should correct the transplant rate for age of dialysis onset.

Page 3 / 12: You state it is widely accepted that transplant within 12 mos for AAV is associated with high risk of recurrence in KTx, but you reference only one publication from 2009.  Can you add others and state the increased risk described for transplanting these patients in less than a year from remission of disease?

Pages 3 to 4 /12:  can you give more specific information, if any is available from the literature, on the range of risk recurrence for ANCA pos, clinical remission patients who undergo KTx as per your references?  The table is not clear on this information.

If possible from the literature, can you create a table with information such as:

Risk of AAV recurrence in KTx within a year with:

No clinical symptoms, ANCA neg

No clinical symptoms, ANCA pos (broken down by c vs p is possible)

Does antibody induction therapy, such as Thymoglobulin or Campath, influence recurrent rates of AAV in KTx?

Is there any evidence in the literature that the use of MPAs are associated with any change in the recurrence rate of AAV compared to older IS protocols with AZA or with no anti-proliferative?

Author Response

We are grateful to Reviewer 2 for his/her valuable comments.

Q1) What evidence is there for waiting one year after the resolution of AAV symptoms?  Do any of your references demonstrate the advantage of waiting one year?

A1) Little et al (ref. 30), in their population of 107 renal transplanted patients, found that the strongest predictor of death was transplantation < 1year post-vasculitis remission on both univariate and multivariate analysis (hazard ratio 2.3, P<0,05). The above sentence has been included in the revised version of the manuscript.

Q2) Pages 2/3 of 12: you state the median age of RRT in AAV renal failure was 70, and then you state this group had a low transplant rate. You should correct the transplant rate for age of dialysis onset. 

A2) Due to the rarity of the disease, it is not possible to sort AAV patients according to their age at the time of enlistment. As such, we could not make a formal comparison between AAV and non-AAV transplant candidates. However, according to the Italian Dialysis and Trasplantation Registry, the proportion of patients receiving a KT is strongly influenced by the age at the time of enlistment. In particular, 20% of candidates aged 70 years or more actually receive a suitable kidney [M. Nordio, A. Limido,  M. Postorino  on behalf of the Italian Dialysis and Transplantation Registry. Present and future of kidney replacement therapy in Italy: the perspective from Italian Dialysis and Transplantation Registry (IDTR) Journal of Nephrology (2020) 33:1195–1200]. Similarly, the French REIN Registry reports that about 10% of the patients aged 70-75 years is successfully transplanted within three years of enlistment [Mathilde Lassalle, Elisabeth Monnet, Carole Ayav, Julien Hogan, Olivier Moranne, Cécile Couchoud, REIN registry 2017 Annual Report Digest of the Renal Epidemiology Information Network (REIN) registry. Transpl Int. 2019 Sep;32(9):892-902]. In the revised version of the manuscript, we have included a  sentence referring to this particular issue.

Q3) Page 3 / 12: You state it is widely accepted that transplant within 12 months for AAV is associated with high risk of recurrence in KTx, but you reference only one publication from 2009.  Can you add others and state the increased risk described for transplanting these patients in less than a year from remission of disease?

A3) The optimal time of transplantation is still debated and prospective data are lacking. Unfortunately, we could not find any other recent paper on the topic. Nevertheless, the paragraph has been re-written to further clarify the message.

Q4) Pages 3 to 4/12:  can you give more specific information, if any is available from the literature, on the range of risk recurrence for ANCA pos, clinical remission patients who undergo KTx as per your references? The table is not clear on this information.

A4) In the study by Nachman et al (ref. 43) no statistically significant differences in relapse rates between patients with or without circulating ANCA at the time of transplantation were detected (P=0,75). Also, no differences in recurrence rates were observed between recipients with specific ANCA subtypes. Marco et al (ref. 32) reported on three AAV patients with post-transplant relapse. All of them showed elevated ANCA levels. GöçeroÄŸlu et al (ref. 50) could not found any association between relapse and ANCA titer or subtype. The above sentences have been included in the revised version of the manuscript. 

Q5) If possible from the literature, can you create a table with information such as: Risk of AAV recurrence in KTx within a year with: No clinical symptoms, ANCA neg, No clinical symptoms, ANCA pos (broken down by c vs p is possible)?

A5) To the best of our knowledge, such data are not available. 

Q6) Does antibody induction therapy, such as Thymoglobulin or Campath, influence recurrent rates of AAV in KTx?

A6) As far as we know, such information is not available.

Q7) Is there any evidence in the literature that the use of MPAs are associated with any change in the recurrence rate of AAV compared to older IS protocols with AZA or with no anti-proliferative?

A7) MMF/MPA has shown mixed results. However, in a recent review (ref. 65), it is suggested to opt for MMF/MPA based immunosuppressive regimens.